# LSMSeg: Unleashing the Power of Large-Scale Models for Open-Vocabulary Semantic Segmentation

## Abstract

Open-vocabulary semantic segmentation requires precise pixel-level alignment of visual and textual representations, leveraging text as a universal reference to address visual disparities across diverse datasets. While prior efforts have primarily focused on enhancing visual representations or alignment models, the contribution of textual representations remains underexplored. Moreover, although CLIP excels at capturing image-level features, its limited capacity for fine-grained pixel-level representation poses a major challenge for semantic segmentation. To address these challenges, we propose LSMSeg that employs large language models (LLMs) to generate enriched text prompts incorporating diverse visual attributes such as color, shape, size, and texture, thereby replacing simplistic templates with semantically rich descriptions. In addition, we propose a Feature Refinement Module that adapts visual features from the Segment Anything Model (SAM) to the CLIP space using a lightweight adapter, followed by a learnable weighting strategy to fuse them with CLIP features, enhancing pixel-to-text alignment. To further reduce computational overhead, we introduce a Category Filtering Module to accelerate training and decrease parameter complexity. Extensive experiments demonstrate that LSMSeg significantly enhances cross-modal alignment and achieves strong performance while maintaining efficiency, offering a robust advancement for open-vocabulary semantic segmentation.

## 1 Introduction

Open-vocabulary semantic segmentation (OVSS) seeks to classify each pixel in an image into its most relevant semantic category from a potentially unbounded set, guided by arbitrary or descriptive text inputs Liang et al. (2023); Xu et al. (2023b). This task relies heavily on pre-trained vision-language foundation models, such as CLIP Radford et al. (2021) and Align Jia et al. (2021), to achieve pixel-level alignment between visual and textual features. However, these foundation models trained on image-level paired datasets primarily capture global context rather than localized semantics, limiting their generalization to pixel-level tasks.

Existing strategies to improve alignment fall into two main categories: (1) **Refine region-level visual-text alignment.** Methods like Ghiasi et al. (2022); Liang et al. (2023); Xu et al. (2022); Jiao et al. (2024) employ a category-agnostic mask generator to derive region-level representations that closely resemble image-level features. However, these methods primarily achieve region-level alignment, incurring substantial computational costs and inefficient memory usage. (2) **Refine pixel-level visual-text alignment.** Other works Shan et al. (2024); Xie et al. (2024); Cho et al. (2024) propose to leverage the extra vision foundation models or feature aggregation to enhance rich pixel-level visual representation, thereby compensating for the spatial deficiencies of CLIP features. These methods complement the spatial limitations of CLIP features, which stem from image-level contrastive training that favors global context over local pixel semantics. Nevertheless, these efforts often overlook a critical component: the quality of textual representations. Simple text prompts like `a photo of a {class name}` often lack the **semantic richness** necessary to resolve fine-grained distinctions. Moreover, CLIP's text encoder can struggle with lexical ambiguities, limiting its discriminative power for fine-grained segmentation tasks. This limitation highlights our argument that the quality of textual representations is equally crucial for achieving precise visual-text alignment in OVSS.

Simplistic prompts fall short in three key aspects: First, they lack the detailed semantic information required for fine-grained segmentation tasks, such as differentiating a flower species based on its intricate petal structure and color. Second, the discriminative power of generated text embeddings depends on the CLIP text encoder, which may fail to distinguish between meanings if there are lexical ambiguities. For instance, the word 'bat' could refer to either 'a flying mammal' or 'a piece of sports equipment used in baseball,' and simply encoding the class name will not be enough to differentiate between these two concepts. Third, they fail to leverage multi-modal information, which is crucial for capturing the nuances of complex categories, thereby limiting the model's adaptability to diverse and fine-grained visual contexts. To

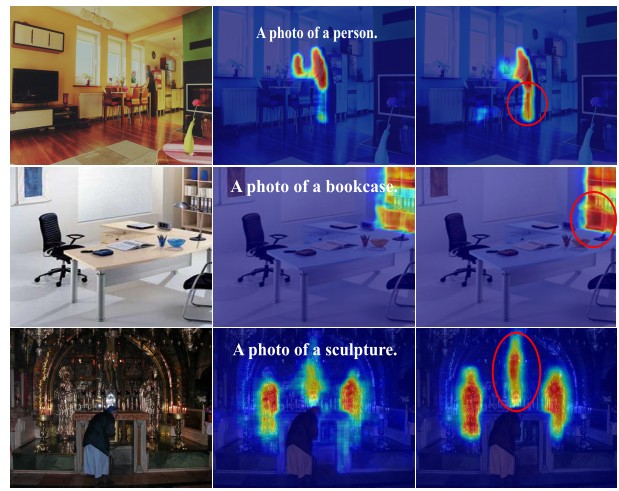

(a) Image     (b) Template Prompt (c) Enriched Prompt

Figure 1: **Visualization of the cost map.** The cost map shows pixel-text alignment, with the first row for the seen class 'person' and the last two for the unseen classes 'bookcase' and 'sculpture'.

solve this problem, we propose to **refine text attribute-level visual-text alignment**. Specifically, we leverage GPT-4 to generate a set of candidate attributes, which are then used to prompt GPT-4 for detailed sentence descriptions tailored to each category. Compared to simple template-based prompts, the attribute-enriched prompts enable the computation of more informative pixel-to-text cost maps Cho et al. (2024), which capture the fine-grained similarity between textual and visual features. This leads to more accurate cross-modal alignment, as shown in Figure 1.

Enhancing textual representations is crucial for capturing attribute-level distinctions in OVSS. However, precise segmentation also depends on high-quality visual features. Prior works Wang et al. (2025); Shao et al. (2024) have demonstrated that intermediate layers of CLIP effectively capture dense features. Motivated by this, we propose a feature refinement module that integrates intermediate CLIP features with a frozen SAM's image encoder Kirillov et al. (2023) and refines the cost map using a Swin-Transformer block and a subsequent linear Transformer block. Meanwhile, we introduce a category filtering module (CFM) to eliminate irrelevant classes and lower computational complexity. By pruning low-relevance categories from the initial cost map, the module improves segmentation accuracy and efficiency. As illustrated in Figure 2, LSMSeg achieves a new state-of-the-art for both efficiency and accuracy. In summary, our main contributions can be summarized as:(1) We propose LSMSeg, a pioneering framework that leverages large language models (LLMs) to create detailed, attribute-enriched text prompts, significantly improving text-visual alignment for OVSS. (2) We propose a feature refinement module by utilizing the precise spatial information of SAM with a category filtering module to reduce computational cost. (3) Extensive experiments across multiple benchmarks demonstrate that LSMSeg achieves state-of-the-art performance in open-vocabulary semantic segmentation.

## 2 RELATED WORK

### 2.1 OPEN-VOCABULARY SEMANTIC SEGMENTATION

Prevalent semantic segmentation methods Chen et al. (2018); Tang et al. (2023); Yuan et al. (2020); Hu et al. (2021); Jin et al. (2021; 2022) are designed for closed sets where only predefined categories can be distinguished, and gathering data for training such models is often costly and time-consuming. As a result, the trend in segmentation tasks is shifting towards open-vocabulary approaches. Existing works primarily focus on two approaches: (1) Refine region-level visual-text alignment. Some works Ghiasi et al. (2022); Liang et al. (2023); Xu et al. (2022); Ding et al.

(2022) adopt a two-stage framework to refine the alignment, where it first trains a class-agnostic mask generator to extract masks and then leverages the pre-trained CLIP to classify each mask. OVSeg Liang et al. (2023) proposes fine-tuning the pre-trained CLIP on these images and building a domain-specific training dataset to address the CLIP's recognition ability on masked background regions. Such a two-stage approach is inefficient and suboptimal, as it uses separate networks for mask generation and classification, lacks contextual information, and incurs high computational costs by requiring CLIP to process multiple image crops. (2) Refine pixel-level visual-text alignment. Unlike two-stage approaches, recent one-stage methods Xu et al. (2023a;b); Liu et al. (2024) directly apply a unified vision-language model for open-vocabulary segmentation. SAN Xu et al. (2023b) attaches a lightweight image encoder to the pre-trained CLIP to generate masks and attention biases. SCAN Liu et al. (2024) proposes a semantic integration module to embed the global semantic understanding and a contextual shift strategy to achieve domain-adapted alignment. CATSeg Cho et al. (2024) introduces a cost aggregation-based framework, incorporating spatial and class aggregation to reason over the multi-modal cost volume effectively. Although these methods demonstrate efficacy, they tend to neglect the pivotal role of language in open-vocabulary semantic seg-

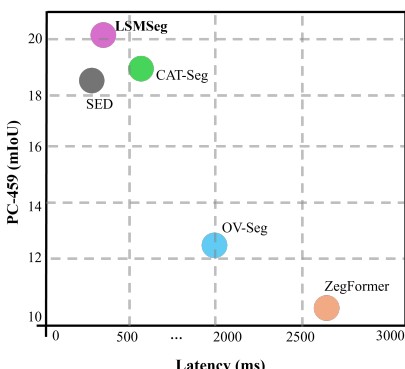

Figure 2: **Segmentation Performance and inference latency on PC-459.** Our LSMSeg outperforms ZegFormer Ding et al. (2022), OV-Seg Liang et al. (2023), CATSeg Cho et al. (2024), and SED Xie et al. (2024), achieving a new state-of-the-art mIoU of 20.3% while maintaining lower latency.

mentation, relying solely on extracting text embeddings from pre-trained vision-language models (VLMs), with almost no works focusing on refining text attributes.

## 2.2 TEXT PROMPT ENHANCEMENT WITH LLMS

Recently, numerous large language models (LLMs), such as GPT Brown et al. (2020) and LLaMA Touvron et al. (2023), have been introduced. Several works Pratt et al. (2023); Khattak et al. (2024); Roth et al. (2023); Roy & Etemad (2023) have showcased their capability to enhance the performance of early vision-language models (VLMs) with LLMs. CuPL Pratt et al. (2023) leverages large language models to produce class-specific prompt descriptions, which are then utilized for text prompt ensembling. WaffleCLIP Roth et al. (2023) uses random descriptors and demonstrates additional improvements by incorporating data-specific concepts generated through LLMs. CoPrompt Roy & Etemad (2023) harnesses a pre-trained large language model's expertise, applying coherence constraints to the text component and data enhancement to the image component to further improve generalization. Some methods Lai et al. (2024) leverage LLMs for reasoning segmentation, highlighting their promise in detailed visual understanding. In this work, we design LLM-generated prompts for OVSS by optimizing attribute selection and combination at the pixel level. This task-specific improvement distinguishes our approach, addressing the unique challenge of aligning fine-grained visual and textual data in segmentation.

## 3 METHOD

### 3.1 PRELIMINARIES

**Problem Definition.** Open-vocabulary semantic segmentation aims to partition an image $I \in \mathbb{R}^{H \times W \times 3}$ into distinct semantic regions based on text descriptions, including classes that were not seen during training. In training, only pixel-level annotations of the seen classes $C_{train}$ are used, with knowledge of their existence and quantity (*i.e.*, how many and what classes are present). Annotations for unseen categories are replaced with an "ignored" label. In an open-vocabulary setting, $C_{test}$ may include new categories that were not encountered during training, meaning $C_{train} \neq C_{test}$. In inference, both seen and unseen classes need to be segmented.

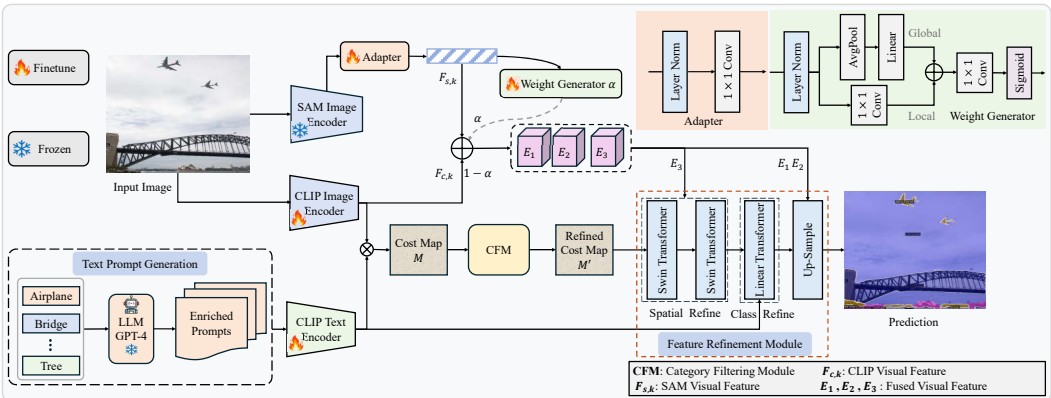

Figure 3: **Overall architecture of our proposed LSMSeg.** We first utilize GPT-4 to generate enhanced text prompts. Next, we propose a category filter module to eliminate irrelevant classes, yielding a refined cost map and reducing computational complexity. Finally, we leverage SAM visual features to address the spatial information deficiency in CLIP visual features through a learnable adapter and weight generator, followed by a feature refinement process to enhance the filtered cost map at both spatial and class levels.

## 3.2 ARCHITECTURE OVERVIEW

Figure 3 illustrates the overall architecture of our proposed LSMSeg, comprising three main components: (a) The *Text Prompts Generation* leverages the GPT-4 model to first select appropriate attributes and then generate descriptive sentences based on those attributes, which are subsequently processed by the CLIP text encoder to obtain text features. (b) The *Category Filter Module* is proposed to reduce computational parameters and accelerate training by filtering irrelevant classes on the pixel-to-text cost map. (c) The *Feature Refinement Module* is introduced to integrate SAM features with CLIP visual features through a learnable weighted fusion strategy to enhance spatial information representation. This fused representation is utilized to refine spatial-level and class-level feature information of the cost map, enabling a more precise and comprehensive feature representation that improves overall model performance.

## 3.3 TEXT PROMPTS GENERATION

To advance open-vocabulary semantic segmentation (OVSS), we introduce a novel approach that leverages GPT-4 to generate enriched text prompts, thereby enhancing pixel-level alignment between textual and visual features. The overall pipeline is illustrated in Figure 4. Firstly, we query GPT-4 with: 'What visual attributes are most relevant for generating descriptive text prompts to enhance pixel-level semantic segmentation?' This yields nine key attributes: color, shape, size, texture, material, position, pattern, action/state, and contextual relationships. Secondly, we further prompt GPT-4 with: 'Describe a {class name} with respect to a given {attribute}. The description should not exceed 77 tokens,

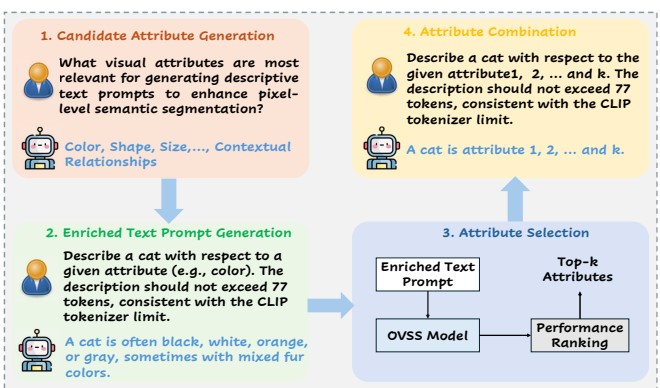

Figure 4: **Comprehensive Linguistic Prompt Generation Pipeline.** The pipeline includes four steps: (1) Candidate Attribute Generation; (2) Enriched Text Prompt Generation; (3) Attribute Selection; and (4) Attribute Combination.

consistent with the CLIP tokenizer limit.' For instance, given the class 'cat' and attribute 'color', GPT-4 generates: 'A cat is often black, white, orange, or gray, sometimes with mixed fur colors.' This procedure yields fine-grained, attribute-specific descriptions to replace simplistic templates such as 'a photo of a {class name}', providing richer textual inputs for the CLIP text encoder. Thirdly, we optimize these prompts by independently assessing the contribution of each attribute without incorporating the Feature Refinement Module. Finally, we integrate the top-k attributes into comprehensive prompts, such as: 'A cat has a small, sleek, and agile shape with a long tail and pointed ears, is small to medium-sized, weighing between 3 to 7 kg, has soft, fluffy fur with a smooth or slightly rough tongue, and is often black, white, orange, or gray.'

### 3.4 CATEGORY FILTERING MODULE (CFM)

Given an input image $I$, we obtain dense visual features $F_c \in \mathbb{R}^{B \times H \times W \times D}$ from the CLIP image encoder, where $B$, $H$, $W$, and $D$ represent the batch size, height, width and channel. Given a set of class names $C$, we leverage LLMs to generate comprehensive linguistic prompts. We obtain text embeddings $T \in \mathbb{R}^{B \times T \times D}$ by feeding these prompts into the CLIP text encoder, where $T$ and $D$ represent the number of class and channel. By computing the cosine similarity between the visual feature $E$ and text embedding $T$, we derive the cost map embedding $M$ as:

$$M_{(i,j,n)} = \frac{F_c(i,j) \cdot T_n}{\|F_c(i,j)\|\|T_n\|}. \tag{1}$$

where $i, j$ denotes the spatial positions, and $n$ indicates the text embedding index. Thus, the cost map embedding $M$ has the dimension of $B \times T \times d \times H \times W$, where $d$ is the channel of the cost map embedding.

To reduce computational overhead and suppress noisy or uninformative text tokens, we apply a top-k token selection when the number of text tokens exceeds a predefined padding threshold $q$. Specifically, we compute the maximum correlation across spatial dimensions and visual prompts:

$$\mathbf{A} = \max_{h,w,d} (\mathbf{M}), \quad \mathbf{A} \in \mathbb{R}^{B \times T}, \tag{2}$$

Then, we select the indices of the top-$k$ highest responding tokens:

$$\mathcal{I}_k = \text{TopK}(\mathbf{A}, \, k = q). \tag{3}$$

These selected token embeddings are gathered and re-normalized:

$$\mathbf{T}' = \text{Gather}(\text{Norm}(\mathbf{T}), \, \mathcal{I}_k), \quad \mathbf{T}' \in \mathbb{R}^{B \times k \times D} \tag{4}$$

where $\text{Norm}(\cdot)$ denotes $\ell_2$ normalization along the feature dimension. $\text{Gather}(\cdot, \mathcal{I})$ retrieves the top-$k$ token embeddings along the token dimension based on the selected indices $\mathcal{I}_k$. For a tensor $\mathbf{T} \in \mathbb{R}^{B \times T \times D}$ and index set $\mathcal{I}_k \in \mathbb{R}^{B \times k}$, this operation selects $k$ tokens per sample in the batch and preserves the prompt and feature structure. Then, we recompute the refined cross-modal cost map via:

$$M'_{(i,j,n)} = \frac{F_c(i,j) \cdot T'_n}{\|F_c(i,j)\|\|T'_n\|}. \tag{5}$$

### 3.5 FEATURE REFINEMENT MODULE

The CLIP Radford et al. (2021) model is trained by image-level contrastive learning and struggles with precisely localizing pixel-level visual features. Its embeddings focus on global visual context instead of the pixel-level semantics within the image. This can be a problem for segmentation, which requires understanding the local context of each pixel about its neighbors. To address this, we further propose leveraging a frozen SAM Kirillov et al. (2023) image encoder to enhance and supplement the spatial information. As shown in Figure 3, we input image $I$ into the SAM image encoder and extract image features $F_s \in \mathbb{R}^{B \times H \times W \times D_s}$ from the last three global attention blocks. A lightweight adapter is proposed to project the SAM features into the same dimensional space as the CLIP features. Next, the Weight Generator generates an adaptive weighting coefficients through

a local and global branch in Figure 3. This weight control the relative contributions of CLIP and SAM features during fusion:

$$E_k = \alpha \times F_{c,k} + (1 - \alpha) \times F_{s,k} \tag{6}$$

where $\alpha$ is the weight generator, k stands for different layers and $E$ is the fused visual feature.

As a segmentation task, it is intuitive to further explore spatial-level and class-level information. We first utilize the Swin-Transformer block Liu et al. (2021) to process the fused visual features for enriching spatial feature information as in Cho et al. (2024); Xie et al. (2024). Then, we perform class-level feature refinement to map textual information onto each pixel, achieving more precise alignment. We feed the text embedding into a linear transformer block generated from the comprehensive prompts through the CLIP text encoder. Finally, we leverage the fused feature again to up-sample the enhanced feature representations. The overall process of feature refinement is summarized as follows:

$$M''_{(i,j,n)} = S\left( [M'_{(i,j,n)}; E_k] \right), \tag{7}$$

$$M'''_{(i,j,n)} = C\left( [M''_{(i,j,n)}; T'_n] \right), \tag{8}$$

$$O = Up\left( [M'''_{(i,j,n)}; E_k] \right), \tag{9}$$

where $S$ and $C$ represent spatial-level and class-level refinements, $E'$ is the intermediate visual feature layer, $Up$ denotes up-sampling, and $O$ is the final prediction.

## 4 EXPERIMENTS

### 4.1 DATASET AND EVALUATION PROTOCOL

We train our model on COCO Stuff Caesar et al. (2018) dataset, and conduct evaluation on ADE20k-847 Zhou et al. (2017) ADE20k-150 Zhou et al. (2017), Pascal Context-459 Mottaghi et al. (2014), Pascal Context-59 Mottaghi et al. (2014), and Pascal VOC Everingham et al. (2010). COCO-Stuff dataset contains 171 annotated classes and includes 118k training, 5k validation, and 41k test images. ADE20K Zhou et al. (2017) is a large-scale benchmark for scene understanding, comprising 20k training images, 2k validation images, and 3k testing images. It includes two sets of annotated classes: ADE20K-150 with 150 classes and ADE20K-847 with 847 classes, although both use the same images. Pascal Context Mottaghi et al. (2014) extends Pascal VOC 2010, offering 4,998 training and 5,005 validation images, with annotations available in two configurations: PC-59 (59 classes) and PC-459 (459 classes). Pascal VOC Mottaghi et al. (2014) consists of 11,185 training images and 1,449 validation images across 20 object classes. Following previous work Cho et al. (2024); Yu et al. (2023); Xu et al. (2023b), Mean Intersection over Union (mIoU) is used as the evaluation metric across all experiments. This metric represents the average intersection-over-union values calculated for each class across all classes.

### 4.2 IMPLEMENTATION DETAILS

Our experiments utilize the pre-trained CLIP model from OpenAI Radford et al. (2021), specifically the ViT-B/16 and ViT-L/14 variants. We fine-tune the CLIP image and text encoder and the total training iteration is set as 80k. The initial learnable fusion weight is empirically set as 0.5 for balance. We use 2 NVIDIA-L40 GPUs for training with a batch size of 4 and the AdamW optimizer with an initial learning rate of $2 \times 10^{-4}$. The weight decay is $1 \times 10^{-4}$ for our model. During training, the input image resolution is $384 \times 384$ for ViT-B/16. For ViT-L/14, the resolution is $336 \times 336$.

### 4.3 COMPARISONS WITH STATE-OF-THE-ART METHODS

We compare our method with existing state-of-the-art approaches across six datasets in Table 1, including the vision-language model (VLM) and training dataset. Apart from SPNet Xian et al. (2019) and ZS3Net Bucher et al. (2019), most methods are developed using VLM as a foundation.

| Method | VLM | Training Dataset | A-847 | PC-459 | A-150 | PC-59 | VOC | VOCb |
|---|---|---|---|---|---|---|---|---|
| SPNet Xian et al. (2019) | - | PASCAL VOC | - | - | - | 24.3 | 18.3 | - |
| ZS3Net Bucher et al. (2019) | - | PASCAL VOC | - | - | - | 19.4 | 38.3 | - |
| Lseg+ Li et al. (2022) | ALIGN EN-B7 | COCO-Stuff | 3.8 | 7.8 | 18.0 | 46.5 | - | - |
| OpenSeg Ghiasi et al. (2022) | ALIGN EN-B7 | COCO Panoptic | 8.1 | 11.5 | 26.4 | 44.8 | | 70.2 |
| ZegFormer Ding et al. (2022) | CLIP ViT-B/16 | COCO-Stuff | 5.6 | 10.4 | 18.0 | 45.5 | 89.5 | 65.5 |
| DeOP Han et al. (2023) | CLIP ViT-B/16 | COCO-Stuff-156 | 7.1 | 9.4 | 22.9 | 48.8 | 91.7 | - |
| OVSeg Liang et al. (2023) | CLIP ViT-B/16 | COCO-Stuff | 7.1 | 11.0 | 24.8 | 53.3 | 92.6 | - |
| SAN Xu et al. (2023b) | CLIP ViT-B/16 | COCO-Stuff | 10.1 | 12.6 | 27.5 | 53.8 | 94.0 | - |
| SCAN Liu et al. (2024) | CLIP ViT-B/16 | COCO-Stuff | 10.8 | 13.2 | 30.8 | 58.4 | **97.0** | - |
| EBSeg Shan et al. (2024) | CLIP ViT-B/16 | COCO-Stuff | 11.1 | 17.3 | 30.0 | 56.7 | 94.6 | - |
| SED Xie et al. (2024) | ConvNeXt-B | COCO-Stuff | 11.4 | 18.6 | 31.6 | 57.3 | 94.4 | - |
| CAT-Seg Cho et al. (2024) | CLIP ViT-B/16 | COCO-Stuff | 12.0 | 19.0 | 31.8 | 57.5 | 94.6 | 77.3 |
| LSMSeg (*ours*) | CLIP ViT-B/16 | COCO-Stuff | **13.1** | **20.3** | **33.2** | **59.7** | 95.4 | **81.1** |
| SimSeg Xu et al. (2022) | CLIP ViT-L/14 | COCO-Stuff | 7.1 | 10.2 | 21.7 | 52.2 | 92.3 | - |
| OVSeg Liang et al. (2023) | CLIP ViT-L/14 | COCO-Stuff | 9.0 | 12.4 | 29.6 | 55.7 | 94.5 | - |
| ODISE Xu et al. (2023a) | CLIP ViT-L/14 | COCO-Stuff | 11.1 | 14.5 | 29.9 | 57.3 | | - |
| SAN Xu et al. (2023b) | CLIP ViT-L/14 | COCO-Stuff | 12.4 | 15.7 | 32.1 | 57.7 | 94.6 | - |
| EBSeg Shan et al. (2024) | CLIP ViT-L/14 | COCO-Stuff | 13.7 | 21.0 | 32.8 | 60.2 | 96.4 | - |
| SCAN Liu et al. (2024) | CLIP ViT-L/14 | COCO-Stuff | 14.0 | 16.7 | 33.5 | 59.3 | 97.2 | - |
| FC-CLIP Yu et al. (2023) | ConvNeXt-L | COCO Panoptic | 14.8 | 18.2 | 34.1 | 58.4 | 95.4 | 81.8 |
| SED Xie et al. (2024) | ConvNeXt-L | COCO-Stuff | 13.9 | 22.6 | 35.2 | 60.6 | 96.1 | - |
| MAFT+ Jiao et al. (2024) | CLIP ViT-L/14 | COCO-Stuff | 15.1 | 21.6 | 36.1 | 59.4 | 96.5 | - |
| DPSeg Zhao et al. (2025) | ConvNeXt-L | COCO-Stuff | 14.9 | 23.5 | 36.4 | 62.0 | **97.4** | - |
| CAT-Seg Cho et al. (2024) | CLIP ViT-L/14 | COCO-Stuff | 16.0 | 23.8 | 37.9 | 63.3 | 97.0 | 82.5 |
| MaskAdapter Li et al. (2025) | ConvNeXt-L | COCO-Stuff | 16.2 | 22.7 | 38.2 | 60.4 | 95.8 | - |
| LSMSeg (*ours*) | CLIP ViT-L/14 | COCO-Stuff | **16.9** | **25.6** | **38.5** | **63.4** | 97.2 | **84.0** |

Table 1: **Comparison with state-of-the-art methods.** We present the mIoU results on six commonly used test sets for open-vocabulary semantic segmentation. The highest results are highlighted in bold, and the second highest are underlined. Compared with other methods, our proposed LSMSeg demonstrates superior performance across all six test sets.

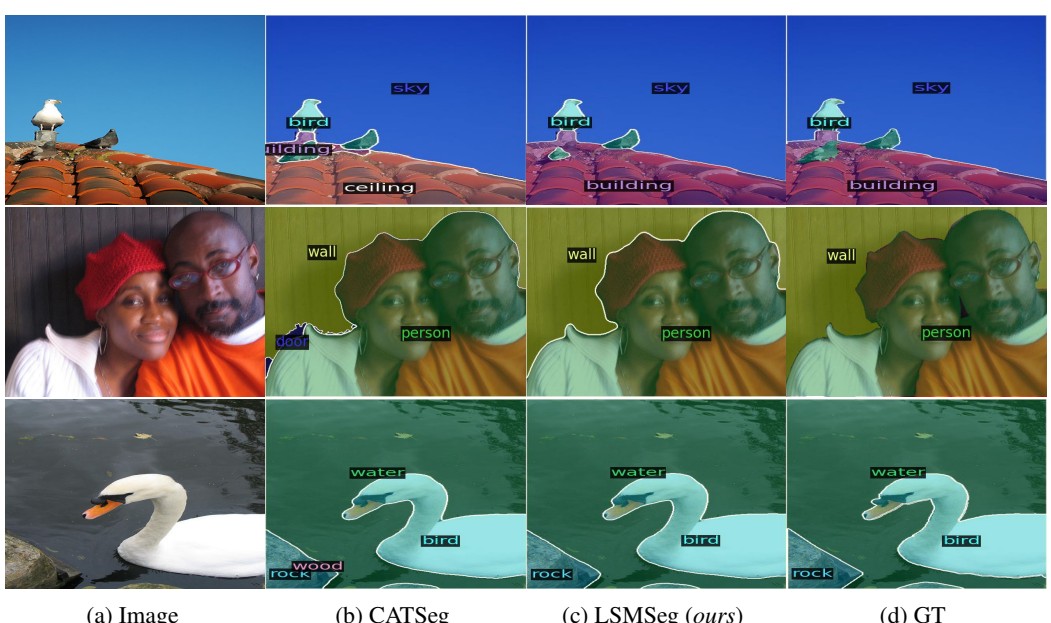

| (a) Image | (b) CATSeg | (c) LSMSeg (*ours*) | (d) GT |
|---|---|---|---|

Figure 5: **Qualitative comparisons on PC-59.** From left to right: input images, results of CAT-Seg, results of our LSMSeg, and ground truth.

To ensure a fair comparison, the results using the same vision-language model are grouped together. Existing open-vocabulary semantic segmentation methods explore various vision–language strategies but still struggle to segment unseen classes accurately. In contrast, our approach achieves notable performance in accurately segmenting both seen and unseen classes. With ViT-B/16 as the vision-language model, our LSMSeg outperforms SAN Xu et al. (2023b), SED Xie et al. (2024), and CATSeg Cho et al. (2024) by 5.7%, 1.6%, and 1.4% on A-150. On PC-459, our method exceed SED Xie et al. (2024), EBSeg Shan et al. (2024), and CAT-Seg Cho et al. (2024) by 1.7%, 3.0%, and 1.3%. When using a larger model ViT-L, our LSMSeg also attains notable performance on all six

datasets. For instance, on ADE-150, our LSMSeg outperforms FC-CLIP Yu et al. (2023), SED Xie et al. (2024) and CAT-Seg Cho et al. (2024) by 4.4%, 3.3% and 0.6%. Our method achieves favorable performance with both base and large models. Additionally, we present qualitative comparisons on PC-59 in Figure 5, demonstrating the superior effectiveness of our proposed LSMSeg approach relative to the cutting-edge method.

## 4.4 ABLATION STUDIES

**Analysis of different prompts.** We have obtained different visual attributes such as color, shape, size, texture, material, position or location, pattern, action or state, and contextual relationship. Here, we utilize ViT-B/16 as the VLM and train on the COCO-Stuff dataset without the feature refinement module. To identify the attributes that play the most crucial role in the segmentation task, we carried out an extensive series of experiments, with the results succinctly summarized in Table 2.

As the performance may vary across different datasets, we present the average results over all datasets to provide a more robust evaluation of the best approach. The baseline method, relying on fixed hand-crafted prompts, achieves an average mIoU of 46.8%. Attributes perform differently: color (47.3%), shape (47.4%), texture (47.4%), and size (47.4%) lead, followed by material (47.0%), pattern (47.0%), position (46.9%), action/state (42.8%), and contextual relationship (39.9%). Experimental results indicate that color, size, shape, and texture are the most influential attributes for generating effective descriptive prompts.

| Methods | A-847 | PC-459 | A-150 | PC-59 | VOC | VOCb | avg. |
|---|---|---|---|---|---|---|---|
| Baseline | 11.0 | 17.9 | 28.4 | 54.6 | 94.6 | 74.3 | 46.8 |
| Color | 11.2 | 17.8 | 28.3 | 55.3 | 94.5 | 76.5 | 47.3 |
| Shape | 11.2 | 18.4 | 28.2 | 54.7 | 94.9 | 77.1 | 47.4 |
| Size | 11.6 | 18.2 | 28.3 | 55.8 | 94.3 | 76.2 | 47.4 |
| Texture | 11.3 | 18.2 | 28.4 | 55.5 | 94.8 | 76.2 | 47.4 |
| Material | 11.0 | 18.2 | 27.7 | 55.9 | 93.6 | 75.4 | 47.0 |
| Positation | 11.4 | 18.0 | 28.0 | 55.0 | 93.1 | 75.7 | 46.9 |
| Pattern | 10.9 | 16.9 | 28.2 | 55.2 | 94.8 | 76.2 | 47.0 |
| Action | 11.2 | 18.1 | 13.9 | 42.2 | 94.9 | 76.4 | 42.8 |
| Context | 7.1 | 17.4 | 16.5 | 29.8 | 94.6 | 73.9 | 39.9 |

Table 2: **Analysis of different prompts.** We conduct an ablation study on each visual attribute individually to verify the positive and negative attributes.

Then, we explore different combinations of these attributes in Table 3. The results show improved average mIoU of 47.5% (size + shape), 47.6% (size + shape + texture), 47.8% (size + shape + texture + color), and 47.5% (size + shape + texture + color + material), respectively. We also query GPT-4 with the prompt: "Describe a {class name} with respect to its typical attributes in one sentence. The description must be under 77 tokens as per the CLIP tokenizer." However, the combinations of well-chosen attributes outperform typical attribute descriptions, highlighting the effectiveness of consistent and strategically selected attributes in enhancing segmentation performance.

| Methods | A-847 | PC-459 | A-150 | PC-59 | VOC | VOCb | avg. |
|---|---|---|---|---|---|---|---|
| Size | 11.6 | 18.2 | 28.3 | 55.8 | 94.3 | 76.2 | 47.4 |
| Size+Shape | 11.3 | 18.2 | 28.2 | 55.3 | 95.1 | 76.8 | 47.5 |
| Size + Shape +Texture | 11.4 | 18.5 | 28.7 | 54.4 | 95.0 | 77.5 | 47.6 |
| Size + Shape +Texture + Color | 11.6 | 18.4 | 28.9 | 55.6 | 94.9 | 77.1 | **47.8** |
| Size + Shape +Texture + Color + Material | 11.6 | 18.3 | 28.7 | 55.4 | 94.9 | 76.3 | 47.5 |
| Typical attributes | 11.3 | 18.2 | 27.9 | 54.6 | 94.8 | 74.6 | 46.9 |

Table 3: **Ablation Study on Attribute Combinations.** We conduct an ablation study on different combinations of different attributes and identify the optimal combination.

**Ablation study for CFM.** We investigate the impact of the filtered class number in CFM without SAM in Table 4. The results show that performance metrics remain remarkably stable across different $k$ values ranging from 16 to 96. Particularly, stability is observed between $k = 32$ and $k = 96$, highlighting the model's robustness to variations in this hyperparameter.

| K | A-847 | PC-459 | A-150 | PC-59 | VOC | VOCb | avg | Latency (ms) |
|---|---|---|---|---|---|---|---|---|
| 16 | 11.7 | 19.0 | 30.9 | 58.3 | 95.0 | 79.9 | 49.1 | 339.5 |
| 32 | 12.8 | 19.7 | 32.1 | 58.0 | 94.8 | 80.0 | 49.6 | **362.8** |
| 48 | 12.6 | 19.6 | 32.0 | 58.1 | 95.2 | 79.9 | 49.6 | 389.1 |
| 64 | 12.6 | 19.9 | 32.0 | 58.4 | 94.8 | 79.7 | 49.6 | 421.7 |
| 96 | 12.6 | 19.8 | 32.2 | 58.4 | 94.8 | 79.6 | 49.6 | 460.2 |

Table 4: **Ablation Study on the Number of Filtered Classes** $k$.

bustness to variations in this hyperparameter. While higher $k$ values lead to a slight increase in latency, the corresponding gains in accuracy beyond $k = 32$ are marginal. Considering the trade-

off between computational efficiency and performance, we therefore select $k = 32$ as the optimal default setting.

**Ablation Study for Feature Refinement Module.** To evaluate the effectiveness of our feature refinement module, we conduct an ablation study and present the results in Table 5. In this experiment, we adopt CLIP ViT-B/16 as the backbone. We independently validate the contributions of spatial refinement and class refinement. We observe that integrating both leads to the optimal results, suggesting their complementary roles in enhancing segmentation.

In addition, we investigate the impact of integrating external visual foundation models. Compared to the baseline without additional features, incorporating SAM (both SAM-B and SAM-L) leads to consistent improvements, demonstrating the benefits of leveraging SAM's strong visual priors. We also evaluate DINOv2-B, which provides moderate gains but remains inferior to SAM-B variants. Notably, combining FRM with SAM yields the best overall performance, confirming that these two components are complementary in strengthening the segmentation ability of our model. This improvement is particularly evident on A-150 and PC-59, where richer prompts and spatial priors strengthen pixel–text alignment for both seen and unseen classes.

| Methods | A-847 | PC-459 | A-150 | PC-59 | VOC | VOCb |
|---|---|---|---|---|---|---|
| LSMSeg(w/o FRM) | 11.6 | 18.4 | 28.9 | 55.6 | 94.9 | 77.1 |
| LSMSeg(w/o Spatial) | 11.6 | 18.5 | 30.5 | 56.8 | 93.2 | 78.6 |
| LSMSeg(w/o Class) | 11.7 | 19.2 | 30.7 | 57.6 | 95.0 | 78.9 |
| LSMSeg(w/ FRM) | **13.1** | **20.3** | **33.2** | **59.7** | **95.4** | **81.1** |
| LSMSeg(w/o SAM) | 12.8 | 19.7 | 32.1 | 58.0 | 94.8 | 80.0 |
| LSMSeg(w/ Dinov2-B) | 12.2 | 18.8 | 31.1 | 57.2 | 94.9 | 78.7 |
| LSMSeg(w/ SAM-B) | 12.5 | 20.3 | 32.1 | 58.8 | 95.2 | 80.2 |
| LSMSeg(w/ SAM-L) | **13.1** | **20.3** | **33.2** | **59.7** | **95.4** | **81.1** |

Table 5: **Ablation study on Feature Refinement Module.** We conduct an ablation study to verify the effectiveness of our proposed Feature Refinement Module (FRM).

| Methods | A-847 | PC-459 | A-150 | PC-59 | VOC | VOCb | avg. |
|---|---|---|---|---|---|---|---|
| mean | 12.9 | 20.1 | 32.8 | 59.3 | 95.0 | 80.8 | 50.2 |
| concat | 13.0 | 20.3 | 32.8 | 59.1 | 95.2 | 80.9 | 50.2 |
| weight generator | **13.1** | **20.3** | **33.2** | **59.7** | **95.4** | **81.1** | **50.5** |

Table 6: **Ablation study on different fusion strategies.** Here, 'mean' denotes the element-wise average of CLIP and SAM features, while 'concat' refers to feature concatenation followed by a projection for dimensional alignment.

We additionally present an ablation study on visual feature fusion strategies in Table 6. Compared with simple averaging (mean) and straightforward concatenation (concat), our proposed weight generator achieves the best performance with an average score of 50.5%. This demonstrates

| Methods | Learnable Params (M) | Training Time (min) | Latency (ms) | GFLOPs |
|---|---|---|---|---|
| ZegFormer | 103.3 | 1148.3 | 2700 | 19,425.6 |
| OVSeg | 102.8 | - | 2000 | 19,345.6 |
| CAT-Seg | 70.3 | 693.7 | 535.2 | 3459.7 |
| LSMSeg(w/o SAM-L) | **70.3** | **446** | **362.8** | **2122.0** |
| LSMSegLSMSeg(w/ SAM-L) | 73.4 | 546 | 426.0 | 3140.6 |

Table 7: **Efficiency comparison**. All results are measured with the Nvidia-L40 GPU.

that adaptively learning fusion weights provides a more effective balance between CLIP and SAM features, leading to consistently improved results across different datasets.

**Model efficiency.** In Table 7, we compare the efficiency of LSMSeg with recent methods in terms of learnable parameters, training time, inference latency, and computational cost (GFLOPs). LSMSeg demonstrates strong efficiency in both training and inference, benefiting from the category filtering module. When combined with SAM-L, LSMSeg maintains competitive efficiency while further improving segmentation performance, highlighting the effectiveness of integrating strong visual information with our lightweight design.

## 5 CONCLUSION

In this work, we introduce LSMSeg, a novel framework that advances open-vocabulary semantic segmentation (OVSS) by effectively modeling the relationship between textual and visual representations. By leveraging GPT-4 to generate attribute-based text prompts, LSMSeg enriches the semantic content of textual inputs, enabling more precise pixel-level alignment with visual features. Furthermore, our Category Filtering Module (CFM) and Feature Refinement Module optimize computational efficiency and segmentation accuracy by pruning irrelevant categories. Lastly, we propose a Feature Refinement Module that dynamically integrates with CLIP visual features to achieve both spatial and class-level feature refinement. Extensive experiments show that LSMSeg not only achieves state-of-the-art performance but also maintains efficient inference with lower latency.

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

## A APPENDIX

### A.1 THE USE OF LARGE LANGUAGE MODELS (LLMS)

In this paper, we propose a novel approach that leverages Large Language Models (GPT-4) to generate attribute-enriched text prompts, enabling more precise alignment between visual and textual representations and achieving significant improvements in open-vocabulary semantic segmentation (OVSS). Specifically, we first generate candidate attributes and systematically validate their effectiveness through extensive experiments. Based on these results, we then select the optimal attribute combinations. Ultimately, this process yields the most effective text prompts for guiding OVSS.

### A.2 ABLATION STUDY ON FINE-TUNING THE ENCODER OF LSMSEG.

Table 8 presents an ablation study on fine-tuning components of CLIP. Due to computational cost constraints, the study did not fine-tune the SAM model, focusing instead on the specified CLIP components. When we freeze the CLIP encoder, the lowest result is achieved across six datasets. The best fine-tuning strategy for CLIP is to fine-tune query and value projections only, with an average result of 50.5%.

| Methods | A-847 | PC-459 | A-150 | PC-59 | VOC | VOCb | avg. |
|---------|-------|--------|-------|-------|-----|------|------|
| Freeze | 8.1 | 13.3 | 25.9 | 46.9 | 83.4 | 61.6 | 39.9 |
| $\text{CLIP}_{qk}$ | 11.6 | 18.2 | 30.7 | 56.2 | 94.7 | 78.5 | 48.3 |
| $\text{CLIP}_{kv}$ | 12.7 | 19.9 | 32.8 | 59.0 | 95.0 | 80.2 | 49.9 |
| $\text{CLIP}_{qv}$ | 13.1 | 20.3 | 33.2 | 59.7 | 95.4 | 81.1 | 50.5 |

Table 8: **Ablation study on fine-tuning the encoder of LSMSeg.** We conduct an ablation study on fine-tuning the CLIP encoder during training. $q, k$, and $v$ of CLIP are query, key, and value projections.

A.3    EXAMPLES OF ATTRIBUTE-ENRICHED TEXT DESCRIPTIONS

In this section, we present some examples of detailed class descriptions generated using the GPT-4 model.

**Generated descriptions for 'bicycle'**

- ```
  A bicycle has a two-wheeled frame with handlebars and a
  seat, is medium-sized at around 1 to 1.5 meters in length,
  has a smooth metal frame, rubber tires, and textured handle
  grips, and is often red, blue, black, or metallic with shiny
  or matte finishes.
  ```

**Generated descriptions for 'car'**

- ```
  A car has a boxy or sleek aerodynamic shape with four
  wheels, varies in size from compact to SUVs and large
  sedans, has a smooth metal body, rubber tires, and leather
  or fabric seats, and is usually white, black, red, or blue
  with a glossy finish.
  ```

**Generated descriptions for 'airplane'**

- ```
  An airplane has a long fuselage with two wings and a tail
  fin, is very large, ranging from small private jets to
  massive airliners, has a smooth metal surface with rivets
  and windows, and is typically white, gray, or silver,
  sometimes with colorful airline logos.
  ```

**Generated descriptions for 'bench'**

- ```
  A bench has a long, rectangular seat with a flat or slightly
  curved surface, is medium to large, seating two to four
  people, has a smooth wooden surface or a textured metal or
  stone finish, and is often brown, gray, or green, blending
  into outdoor environments.
  ```

A.4    MORE QUALITATIVE RESULTS

We show more qualitative comparisons on PC-459 and ADE-150 in Figure 6 and 7.

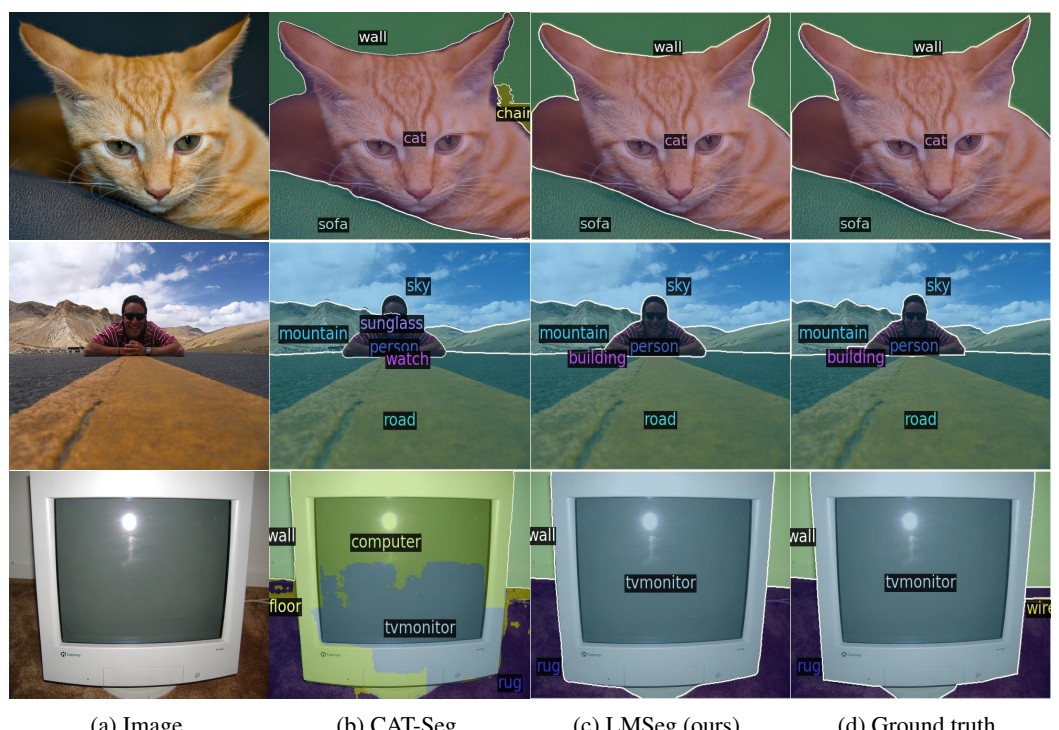

|  (a) Image | (b) CAT-Seg | (c) LMSeg (ours) | (d) Ground truth |

Figure 6: **Qualitative comparisons on PC-459.** From left to right: input images, results of CAT-Seg, results of our LMSeg, and ground truth.

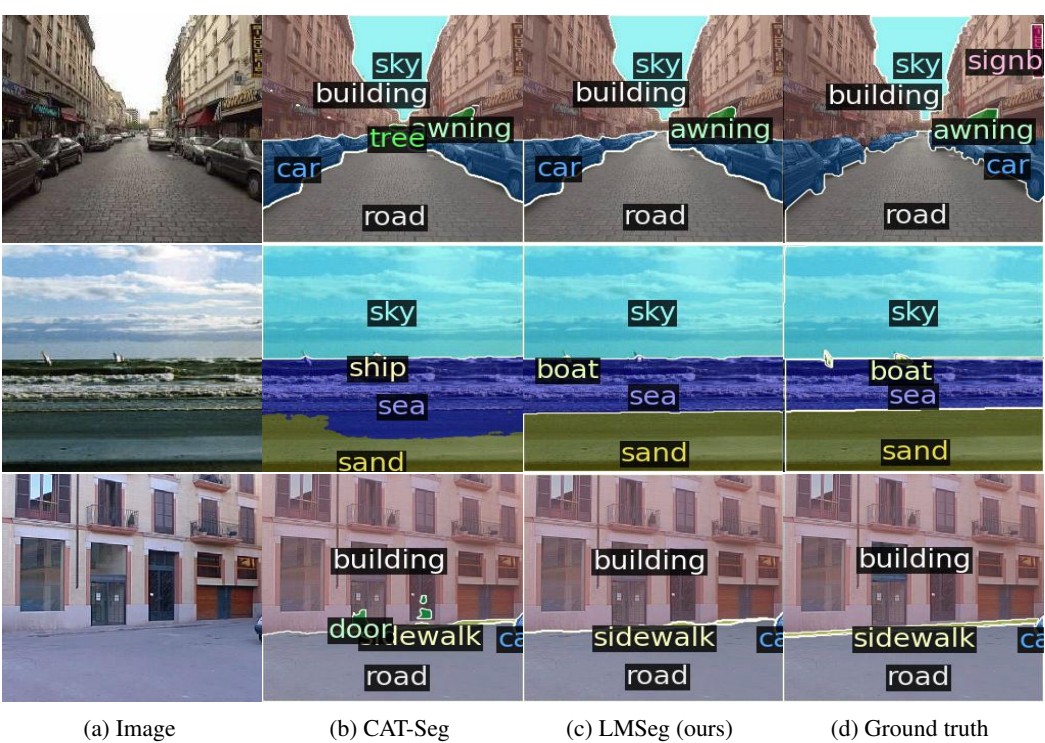

|  (a) Image | (b) CAT-Seg | (c) LMSeg (ours) | (d) Ground truth |

Figure 7: **Qualitative comparisons on A-150.** From left to right: input images, results of CAT-Seg, results of our LMSeg, and ground truth.

