# OpenReview forum: "LSMSeg: Unleashing the Power of Large-Scale Models for Open-Vocabulary Semantic Segmentation"
_ICLR.cc/2026/Conference — ICLR 2026 Conference Withdrawn Submission_

### Official Review · Reviewer_54dY · 2025-10-20

**Soundness:** 3
**Presentation:** 3
**Contribution:** 1
**Rating:** 2
**Confidence:** 5

**Summary:**

This paper enhances expressiveness of the text prompts by levering the LLM, thereby enhancing the performance in 2D open-vocabulary semantic segmentation.

**Strengths:**

* The motivation of this paper is clearly explained. Paper categorizes the previous methods into two classes and points out the limitations of them. The method suggested by the paper leads to performance improvement compared to the state-of-the-art methods.
* Paper presents experiment results on both effectiveness and efficiency, demonstrating the thorough evaluation compared to the previous methods.

**Weaknesses:**

My main concern of this paper is lack of novelty. Most of the components explained in this paper have already been widely used in the 2D/3D scene understanding.

* **Text prompt generation:** Paper augments the text-prompt by using GPT-4 and considering the multiple attributes such as color, shape, texture and etc. These augmented texts improve the open-vocabulary capabilities by allowing the model to learn more diverse vocabularies. However, using GPT-augmented captions to improve semantic generalizabilities of the model has been widely explored in 2D/3D scene understanding community [a, b, c, d] which lacks the technical novelty.

* **Category Filtering Module:** CFM module prunes the unrelated texts based on the cosine similarity scores. However, similar technique is already used in MaskCLIP [e] which is named as *prompt denoising*. Furthermore, pruning the unrelated classes is simple trick to improve the computational efficiency of the model which can't be regarded as major contribution.

* **Feature Refinement Module:** FRM module predicts the adaptive weights to better fuse CLIP and SAM features with additional learnable module. However, interpolating two features with predicted weights is also widely used in the computer vision community [f] and also seems minor technical trick.

I also have some minor concerns/questions.

* In the *Introduction* section, the paper introduces three limitations of the previous methods. However, it is hard to agree to third argument. For example, CAT-Seg [g] proposes cost-aggregation method to refine the relevance scores between visual features and text features which are multi-modal information. I think paper needs to specify this argument more concretely.
* Typo in the last row of Tab. 7.


---

**References**

[a] Huang, Yuantian, Satoshi Iizuka, and Kazuhiro Fukui. "Training-Free Zero-Shot Semantic Segmentation with LLM Refinement." BMVC. 2024.

[b] Sun, Wenfang, et al. "Training-free semantic segmentation via llm-supervision." arXiv preprint arXiv:2404.00701 (2024).

[c] Jia, Baoxiong, et al. "Sceneverse: Scaling 3d vision-language learning for grounded scene understanding." European Conference on Computer Vision. Cham: Springer Nature Switzerland, 2024.

[d] Kong, Lingdong, et al. "Talk2Event: Grounded Understanding of Dynamic Scenes from Event Cameras." arXiv preprint arXiv:2507.17664 (2025).

[e] Zhou, Chong, Chen Change Loy, and Bo Dai. "Extract free dense labels from clip." European conference on computer vision. Cham: Springer Nature Switzerland, 2022.

[f] Wei, Kan, et al. "MGFNet: An MLP-dominated gated fusion network for semantic segmentation of high-resolution multi-modal remote sensing images." International Journal of Applied Earth Observation and Geoinformation 135 (2024): 104241.

[g] Seokju Cho, Heeseong Shin, Sunghwan Hong, Anurag Arnab, Paul Hongsuck Seo, and Seungryong Kim. Cat-seg: Cost aggregation for open-vocabulary semantic segmentation. In Proceedings of the IEEE/CVF Conference on Computer Vision and Pattern Recognition, pp. 4113–4123, 2024.

**Questions:**

NA

---

### Official Review · Reviewer_J4He · 2025-10-29

**Soundness:** 2
**Presentation:** 3
**Contribution:** 2
**Rating:** 4
**Confidence:** 4

**Summary:**

This paper introduces a novel open-vocabulary semantic segmentation model, LSMSeg, which advances the field from three key perspectives: 1) leveraging large language models (LLMs) to generate enriched and context-aware text prompts, thereby enhancing textual representations; 2) proposing a feature refinement module that effectively adapts visual features derived from the Segment Anything Model (SAM); and 3) introducing a category filtering module to improve computational efficiency.
Extensive experiments on six widely used benchmarks demonstrate that LSMSeg outperforms existing state-of-the-art methods, achieving superior performance and exhibiting high efficiency during both training and inference.

**Strengths:**

**Quality:** The empirical quality of this work is strong. The authors validate their approach by achieving new state-of-the-art (SOTA) performance across six challenging benchmark datasets. The experimental section is comprehensive, and the ablation studies are carefully designed to justify the contribution and effectiveness of each component within the proposed architecture.

**Clarity:** The paper is well-written and clearly structured. The proposed method is broken down into three logical components: Text Prompts Generation, Category Filter Module, and Feature Refinement Module, each explained in its own subsection. The architecture is supported by an excellent high-level diagram (Figure 3) and a specific, helpful diagram for the novel prompt generation pipeline (Figure 4).

**Weaknesses:**

**Originality:** While the proposed Category Filtering Module (CFM) demonstrates a certain degree of novelty within the field, the text prompt generation technique largely follows established practices in prompt learning. Moreover, the spatial refinement module closely resembles the mask decoder used in CAT-Seg, and the integration of SAM’s visual features is implemented through a straightforward weighted summation. Therefore, the proposed method can be considered incremental rather than fundamentally innovative.

**Significance:** As the techniques employed in this work are largely incremental, the performance improvements over existing methods are moderate. While the results are consistent and demonstrate solid empirical validation, the overall advancement in methodology and performance impact remains limited.

**Questions:**

- In Figure 1, the authors should provide a clearer illustration of the enriched text prompts used in the examples to help readers better understand their composition and role in the framework.
- In Table 1, the authors are encouraged to include more recent comparison methods (2025) when using CLIP ViT-B/16 as the vision-language model (VLM), to ensure a fair and up-to-date performance comparison.
- The process for selecting the best prompts seems computationally expensive and complex. It involves (1) generating candidate attributes, (2) generating descriptions for each, and (3) running independent training experiments to rank the performance of each attribute individually (as shown in Table 2) before combining the best ones.

---

### Official Review · Reviewer_vYEi · 2025-10-30

**Soundness:** 3
**Presentation:** 3
**Contribution:** 2
**Rating:** 4
**Confidence:** 5

**Summary:**

This paper proposes LSMSeg, a one-stage open-vocabulary semantic segmentation model that fuses CLIP and SAM features and enhances the richness of text prompts with diverse visual attributes using via LLM. The method achieves strong performance on standard benchmarks like ADE20K and Pascal Context, outperforming prior methods in several settings.

**Strengths:**

- The feature refinement module, which fuses SAM and CLIP features via a learnable fusion and token filtering mechanism, is an interesting and well-motivated architectural design.

- The paper presents consistent improvements across standard datasets, supported by ablations on prompt generation, feature fusion, and token filtering.

- Qualitative results demonstrate robustness on challenging scenes involving multiple objects and small, fine-grained categories.

**Weaknesses:**

- The paper’s contributions overlap substantially with those of the CVPR 2024 paper “USE: Universal Segment Embeddings for Open-Vocabulary Image Segmentation.” Both LSMSeg and USE use attribute-rich text prompts generated by large language models (LLMs) to enhance open-vocabulary segmentation, and both aggregate intermediate features from CLIP (across layers) to improve visual representation. These are two of the main innovations claimed by LSMSeg, yet the paper does not cite or discuss USE, weakening the novelty claim and positioning.

- Ablation studies are limited in scope. The paper does not analyze the sensitivity to token filtering granularity, fusion configurations, or CLIP/SAM layer selection.

- Inference efficiency is not reported. Given the use of dual visual encoders (SAM and CLIP) and additional fusion modules, the computational cost remains unclear.

- The choice of SAM’s image encoder is underexplained. While it is said to provide structural priors, the paper does not compare it against more popular backbones like DINOv2 or justify its selection with empirical evidence or prior benchmarks.

**Questions:**

- How much does the segmentation performance depend on the level of detail or specificity in the LLM-generated text prompts? For example, do longer or more descriptive prompts yield better results than shorter or generic ones?

- How much latency or overhead does the fusion module introduce at inference time?

- Why was SAM chosen over more common visual backbones like DINOv2?

---

### Official Review · Reviewer_TUSV · 2025-10-31

**Soundness:** 3
**Presentation:** 3
**Contribution:** 2
**Rating:** 4
**Confidence:** 5

**Summary:**

Open-vocabulary semantic segmentation (OVSS) relies on vision-language models (VLMs) for pixel-level alignment, but prior works neglect textual representation quality and CLIP’s fine-grained pixel limitation. LSMSeg addresses these issues by using GPT-4 to generate attribute-enriched text prompts (color, shape, etc.), designing a Feature Refinement Module to fuse SAM and CLIP features via a lightweight adapter, and introducing a Category Filtering Module to reduce computation. Extensive experiments on 6 datasets (e.g., PC-459 mIoU=20.3%) show it achieves state-of-the-art (SOTA) performance with high efficiency. Its contributions include pioneering LLM-based attribute prompts for OVSS, fusing SAM-CLIP features efficiently, and balancing performance/efficiency with the filtering module .

**Strengths:**

1. Originality in Text Enhancement: It is one of the first works to systematically use LLMs (GPT-4) to generate multi-attribute text prompts for OVSS, addressing the long-neglected textual representation gap and demonstrating that attributes like color/shape significantly improve pixel-text alignment (avg. mIoU up 1% vs. baseline) .
2. Balanced Performance & Efficiency: The Category Filtering Module effectively reduces computation (latency 362.8ms for k=32 vs. CATSeg’s 535.2ms) without accuracy loss, and the Feature Refinement Module successfully complements CLIP’s spatial defects with SAM, achieving SOTA on 6 datasets .
3. Clear Experimental Design: The experimental setup is rigorous, with systematic comparisons against 18 SOTA methods, ablation studies for all core modules (text prompts, feature fusion, filtering), and evaluation across diverse datasets (ADE20k, Pascal Context), ensuring result reliability .

**Weaknesses:**

1. Incomplete Baseline Comparison: It fails to compare with mainstream VLM+SAM pipelines (e.g., Qwen-VL+SAM), which are widely used in OVSS for their API-based deployment and high precision. Without this comparison, the practical value of LSMSeg (vs. low-cost, high-performance VLM+SAM) is unclear .
2. Superficial LLM Integration: LLM is only used for offline attribute generation, not integrated into the model training loop (e.g., no feedback from segmentation results to optimize prompts). This contrasts with advanced works (e.g., LLMSeg) that use LLMs for end-to-end reasoning, limiting innovation .
3. Insufficient Principle Analysis: The Feature Refinement Module’s weight generator (adaptive fusion of SAM/CLIP) lacks mathematical derivation or visualization of weight distribution, and the Category Filtering Module does not explain why k=32 is optimal beyond empirical results, weakening methodological depth .

**Questions:**

1. Could you add experiments comparing LSMSeg with VLM+SAM pipelines (e.g., Qwen-VL+SAM via API) in terms of precision (mIoU), latency, and deployment cost? This is critical to validate LSMSeg’s practical advantages.
2. How does GPT-4’s prompt design (e.g., query wording for attributes) affect prompt quality? Could you provide examples of failed prompts (e.g., incorrect attributes for rare classes) and analyze their impact on segmentation?
3. Why not integrate LLM into the training loop (e.g., using LLM-generated prompts as dynamic supervision) instead of offline generation? Would this improve the model’s adaptability to unseen classes?
4. For the Feature Refinement Module, could you visualize the weight distribution of SAM/CLIP features in different regions (e.g., object edges vs. backgrounds) to verify the adaptive fusion mechanism’s effectiveness?

---

### Note · Authors · 2025-11-15

I have read and agree with the venue's withdrawal policy on behalf of myself and my co-authors.